# Optimal Insertion Depth of Gastric Decompression Tube with a Thermistor for Patients Undergoing Laparoscopic Surgery in Trendelenburg Position

**DOI:** 10.3390/ijerph192214708

**Published:** 2022-11-09

**Authors:** Hwa Song Jong, Tae Won Lim, Ki Tae Jung

**Affiliations:** 1Department of Anesthesiology and Pain Medicine, Chosun University Hospital, Gwangju 61453, Korea; 2Department of Anesthesiology and Pain Medicine, College of Medicine and Medical School, Chosun University, Gwangju 61452, Korea

**Keywords:** anatomic landmarks, body temperature, decompression, intubation, gastrointestinal, monitoring, intraoperative, regression analysis

## Abstract

Monitoring core temperature is crucial for maintaining normothermia during general anesthesia. Insertion of a gastric decompression tube (GDT) may be required during laparoscopic surgery. Recently, a newly designed GDT with a thermistor for monitoring esophageal temperature has been introduced. The purpose of the present study was to evaluate the optimal insertion depth of a GDT with a thermistor. Forty-eight patients undergoing elective laparoscopic surgery in the Trendelenburg position were included in the study. The GDT was inserted to a depth of nose–earlobe–xiphoid distance (NEX) + 12 cm and withdrawn sequentially, 2 cm at a time, at 5-min intervals. Temperatures of the GDT thermistor were compared with the core temperature of the tympanic membrane (TM) using Bland and Altman analysis. The correlation between optimal insertion depth of the GDT and anatomical distance (cricoid cartilage to the carina, CCD; carina to the left hemidiaphragm, CLHD) was evaluated, and a mathematical model to predict the optimal insertion depth of the GDT with a thermistor was calculated. Temperatures of TM and GDT thermistor at NEX + 4 cm showed good agreement and strong correlation, but better agreement and stronger correlation were seen at the actual location with the most minor temperature differences. The optimal insertion depth of the GDT was estimated as −15.524 + 0.414 × CCD − 0.145 × CLHD and showed a strong correlation with the actual GDT insertion depth (correlation coefficient 0.797, adjusted R^2^ = 0.636). The mathematical formula using CCD and CLHD would be helpful in determining the optimal insertion depth of a GDT with a thermistor.

## 1. Introduction

General anesthesia impairs the thermoregulatory responses and vasoconstriction threshold, leading to interferences in maintaining core temperature [1]. Moreover, surgical patients are exposed to cold environments [2]. Thus, patients undergoing general anesthesia develop hypothermia easily and have a risk of complications such as coagulation disorder, infection, heart complications, or delayed wound healing [1,3,4,5,6]. Therefore, avoiding hypothermia is very important for preventing complications in patients undergoing surgery [2,3,7]. Although it is essential to use various methods to prevent intraoperative hypothermia, accurate monitoring of core body temperature is also necessary [3,4]. Core body temperatures can be monitored at multiple locations, such as the tympanic membrane (TM), pulmonary artery, esophagus, and nasopharynx [3,4,5]. Use of an esophageal temperature probe is considered to be a particularly effective and accurate method for continuous measurement of core temperature [5,6].

In recent studies, it has been shown that an NGT is not routinely required, does not reduce postoperative complications, and rather, increases patient discomfort and pulmonary complications [8,9,10,11]. However, using a nasogastric tube (NGT) may sometimes be necessary for several reasons. During various laparoscopic surgery procedures, gastric decompression using an NGT or orogastric tube may be required to avoid intraabdominal organ injury or enhance the surgical view [12,13,14]. Patients at risk of potential complications, such as pulmonary aspiration due to regurgitation of gastric contents, may require gastric decompression during laparoscopic surgery [15,16]. Especially, gastric decompression tube (GDT) insertion is essential when an ileus or a full stomach is suspected [17]. Moreover, the Trendelenburg position can increase intraabdominal and intragastric pressure, leading to increased stomach fluid aspiration [13,16]. Therefore, although the GDT should not be inserted as a routine procedure, sometimes it may be required for the surgical procedure or prevention of potential complications.

Recently, a newly designed GDT with a thermistor that can be used not only as a nasogastric tube to facilitate gastric decompression for laparoscopic surgery but also as an esophageal temperature-monitoring device has been introduced. The thermistor is located 35 cm from the end of the GDT; therefore, it is designed so that the thermistor can be located in the esophagus when side-holes are located in the stomach. As it is known, the optimal position for the thermistor of the esophageal stethoscope is about 45 cm from the nose, to place it in the lower 1/3 of the esophagus [18]. Additionally, a method of also inserting the GDT about 12 to 16 cm from the location where the heart and breath sounds can be heard best is used to determine the location of the thermistor of the esophageal stethoscope [18]. However, the new GDT with a thermistor cannot be connected to the stethoscope, and heart or breath sounds cannot be heard; thus, a new evaluation of optimal insertion depth for this device is required.

For this reason, the purpose of the current study was to evaluate the desired insertion depth of a GDT with a thermistor and develop a mathematical model to predict the optimal insertion depth of the GDT based on the patient’s characteristics and the anatomical landmarks of the CXR.

## 2. Materials and Methods

We conducted the study after approval of the Institutional Review Board of Chosun University Hospital (2020-11-024) following the Helsinki Declaration of Ethical Principles for Medical Research Involving Human Subjects in 1975 (revised 2013). A total of 50 patients undergoing elective laparoscopic surgery in the Trendelenburg position, scheduled to last more than 120 min, were assessed for eligibility. The patients with ASA class 1–2 (20–65 years old) were enrolled for the study. We excluded patients with the following conditions: anatomical abnormality of the upper airway, risk of a difficult airway, abnormal central anatomical structures such as airway, diaphragm, or spine on the chest X-ray (CXR), history of disease of or surgery on the stomach or esophagus, risks of bleeding or coagulopathy per the preoperative laboratory results, obese patients with body mass index (BMI) over 30, contraindication to insertion of a GDT for the surgery, and patients who did not take a standing chest PA X-ray before surgery. After a careful explanation, all patients agreed to participate in the study, and written informed consent was obtained. 

The anesthesia in all patients was performed using standard hospital procedure techniques. All patients fasted overnight and were transferred to the operating room after administration of intramuscular midazolam (0.05 mg/kg). Basal monitoring devices (Carescape; GE Healthcare, Milwaukee, WI, USA) such as electrocardiogram, non-invasive blood pressure, pulse oximetry, neuromuscular monitoring sensor, and bispectral index were applied and prewarmed with a forced air-warming device (3M™ Bair Hugger™ Temperature Management Unit Model 505, Arizant Healthcare Inc., A 3M company, Eden Prairie, MN, USA) which was set to a high level (43 °C). A skilled anesthesiologist managed general anesthesia, and endotracheal intubation was performed using video-laryngoscopy (Glidescope^®^ system, Saturn Biomedical Systems, Burnaby, BC, Canada). After the induction of anesthesia, patient warming was maintained during the surgery using a forced air-warming blanket covering the patient’s upper body except for the head and neck. The ambient temperature was maintained at approximately 22–24 °C and monitored with an indoor thermometer (SH-104S, Saehan, Busan, Korea) near the patient’s head. 

Core temperatures were measured with a tympanic thermometer (Thermoscan IRT 4020, Braun, Kronberg, Germany). When the TM temperature was constant after measuring three consecutive times at 10-s intervals, the temperature was assumed as the core body temperature. The GDT with a temperature probe (ST probe, S&S MED, Gunpo, Korea, Figure 1A) sized 18 Fr. (total length of 80 cm) was inserted carefully by the trained anesthesiologist and confirmed by ultrasound (M-Turbo^®^, Fujifilm Sonosite, Inc, Bothell, WA, USA) and video-laryngoscopy [19]. The reference insertion length of the GDT was estimated for each patient using the conventional nose–earlobe–xiphoid distance (NEX) method [20,21]. Initially, the GDT was inserted from the nose to a depth of NEX + 12 cm. However, if the NEX + 12 cm was longer than 80 cm, the initial insertion depth was started from 80 cm because the total length of the GDT was 80 cm. Finally, each probe was connected to the clinical monitor and recorded temperatures.

Thirty minutes after Trendelenburg positioning of patients for the surgery and initiation of pneumoperitoneum, when the changes in the body temperatures stabilized, initial temperatures of the TM and GDT thermistor were assessed. After 5 min, the GDT was withdrawn 2 cm, and the temperatures of each site were measured when the temperature change of the GDT thermistor stabilized below 0.1 °C. Eventually, the depth of the GDT was changed from NEX + 12 to NEX − 12 cm with a 2-cm interval, and the temperature at each position was measured. At the same time, the core body temperatures of the TM were measured and compared with the temperature of the GDT thermistor. For the further evaluation of the optimal insertion depth of the GDT, anatomical landmarks, such as distance from the cricoid cartilage to the carina (CCD) and distance from the carina to the left hemidiaphragm (CLHD), were measured using an electronic caliper on a CXR of each patient (Figure 1). Since the cricoid cartilage is located at the level of the C6 spinal vertebrae [22], the inferior border of the spinous process of the C6 vertebrae was assumed to be the level of the cricoid cartilage. 

The primary outcome of the study was to evaluate the desired insertion depth of a GDT with a thermistor. To evaluate the desired insertion depth, we assessed the accuracy of temperatures of the GDT thermistor at each location according to the NEX method. We also compared the correlation between core temperature and temperatures at the desired insertion depth and the actual insertion depth of the GDT to evaluate the optimal insertion depth for further analysis. The secondary outcome of the study was to assess the correlation between the characteristics of patients, including gender, height, weight, BMI, CCD, or CLHD, and the actual insertion depth of the GDT that showed the most minor difference in temperature between the TM and the GDT thermistor or desired insertion depth using the NEX method. With those results, we calculated a mathematical model to predict the optimal insertion depth of GDT with a thermistor.

### Statistical Analysis

The sample size was predetermined based on the results of the previous study [3]. The correlation coefficient between the insertion depth and the difference in temperature with a 95% confidence interval (95% CI) was calculated as 0.4543. It was identified that 36 patients were required with α = 0.05 and a power of 0.85 (β = 0.15). Considering dropouts, the total sample size was designed to enroll 50 patients. Statistical analyses were performed using IBM SPSS Statistics (version 28.0, IBM Corp., Armonk, NY, USA).

The accuracy of the GDT thermistor at each location was estimated as the difference between the esophageal temperature using the GDT thermistor and the core body temperature at the TM. Clinically acceptable accuracy of the GDT thermistor was assumed as a mean difference of ±1.96 standard deviations (SD) in temperatures between −0.5 °C and 0.5 °C, which equals the Bland–Altman limits of agreement [3]. Therefore, the desired insertion depth for the GDT was considered the location with the best Bland–Altman limits of agreement and correlation. The actual location was regarded as the depth with the most minor temperature differences between the TM and the GDT thermistor. Bland–Altman analysis was performed to evaluate the difference between the GDT thermistor’s temperatures and TM’s core temperatures. The correlation between the desired or actual insertion depth of the GDT thermistor and patient characteristics, such as gender, age, height, weight, BMI, CCD, and CLHD were analyzed using simple linear regression. Then, the regression equation for the optimal insertion depth of GDT was calculated by multivariate backward stepwise regression using variables with a statistical significance (*p*-value < 0.05 according to Pearson’s correlation test) and a strong correlation. Multicollinearity between variables was verified using an appropriate variation inflation factor of less than 5. The difference and correlation between optimal and actual insertion depth of the GDT were evaluated by Bland–Altman analysis and linear regression. 

## 3. Results

A total of 50 patients scheduled for elective laparoscopic surgery in the Trendelenburg position were assessed for eligibility and met inclusion criteria. All patients agreed to participate and were enrolled. Two patients were excluded because of conversion to open abdomen surgery. Finally, 48 patients were analyzed.

Table 1 shows the characteristics of patients, including the distance of thoracic structures obtained from the CXR. The mean and SD of GDT thermistor temperature and the differences in temperatures between the TM and GDT thermistor according to the insertion depths and locations with the most minor temperature differences are presented in Table 2. 

The mean difference ±1.96 SD of temperature was between −0.5 °C and 0.5 °C at depths of NEX + 8 cm to NEX + 2 cm between the TM and the GDT thermistor (Figure 2). At a depth of NEX + 4 cm, the mean differences in temperature were the most minor between the GDT thermistor and the TM. Bland–Altman analysis of TM temperature versus GDT thermistor temperature at NEX + 4 cm showed good agreement without inclination. The differences in the temperature were distributed within ±2 SD except for one value with an average bias of 0.177 (95% CI 0.105 to 0.249) with a limit of agreement of −0.307 to 0.661 (Figure 3A). However, one point was out of the clinically acceptable range (±0.5 °C). Linear analysis showed a strong correlation coefficient of 0.862 between TM temperature and GDT thermistor temperature at the NEX + 4 cm, with adjusted R^2^ = 0.738 and RMSE (root mean square error) = 0.238 (*p* < 0.001, Figure 3B). 

Nevertheless, Bland–Altman analysis of temperature between TM and GDT thermistor at the actual location with the most minor temperature differences showed better agreement than those at a depth of NEX + 4 cm. The temperature differences were distributed within ±2 SD with an average bias of −0.148 (95% CI −0.195 to 0.100) with a limit of agreement of −0.469 to 0.173 (Figure 3C). All points were within the clinically acceptable range (±0.5 °C). There was a stronger correlation in temperature between TM and GDT thermistor at the actual location with a correlation coefficient of 0.943 (adjusted R^2^ = 0.889, RMSE = 0.162, *p* < 0.001, Figure 3D). Thus, we assumed that the optimal insertion depth of the GDT is the actual depth at the location with the most minor temperature difference between the TM and the GDT thermistor. The average of actual GDT insertion depth was 61.00 ± 5.40 cm (ranging from 52 to 72 cm). The average of differences in the temperatures between TM and GDT thermistor at the actual location with the most minor temperature difference was 0.15 ± 0.16 °C (range −0.10 to 0.50 °C).

We analyzed the correlation between the actual insertion depth of the GDT and the patient’s characteristics (Table 3). Simple linear regression analysis between the actual insertion depth of the GDT and patient characteristics such as height, CCD, and CLHD showed moderate correlations (adjusted R^2^ = 0.605, Durbin–Watson = 1.873, *p* < 0.001). Stepwise multiple regression analysis showed that CCD and CLHD were associated with actual GDT insertion depth. Finally, the optimal GDT insertion depth using CCD and CLHD was calculated as −15.524 + 0.414 × CCD − 0.145 × CLHD. Bland–Altman analysis of actual GDT insertion depth versus optimal GDT insertion depth showed an average bias of −0.06 (95% CI −1.085 to 0.956) with a limit of agreement of −6.979 to 6.858 and negative correlation (r = −0.355, 95% CI −0.581 to −0.079, Figure 4A). The differences in the depths were distributed within ±1.96 SD except for two values. Linear regression analysis between the actual GDT insertion depth versus optimal GDT insertion depth showed a strong correlation coefficient of 0.797 with adjusted R^2^ = 0.636 and RMSE = 3.57 (*p* < 0.001, Figure 4B). The calculated optimal GDT insertion depths were distributed within 95% prediction lines except for two values. 

## 4. Discussion

This study analyzed the temperature differences between TM and GDT thermistor according to the insertion depths. We found that (1) The mean differences ±1.96 SD of temperature between TM and GDT thermistor were the most minor at NEX + 4 cm and showed good agreement and strong correlation (correlation coefficient of 0.862, adjusted R^2^ = 0.738). (2) However, temperature readings between TM and GDT thermistor at the actual location with the most minor temperature differences showed better agreement and stronger correlation than those at a depth of NEX + 4 cm (correlation coefficient of 0.943, adjusted R^2^ = 0.889). (3) There was a strong correlation between the actual insertion depth of the GDT and landmarks such as CCD and CLHD (adjusted R^2^ = 0.769). The optimal insertion depth of GDT was estimated as −15.524 + 0.414 × CCD − 0.145 × CLHD. (4) The actual GDT insertion depth and optimal GDT insertion depth showed a strong correlation, with a correlation coefficient of 0.797 (adjusted R^2^ = 0.636). 

The insertion of a NGT has been used as management in abdominal surgery to facilitate surgical processes by decompressing the distended stomach and enhancing the visibility of the surgical field [9,12,13,14]. However, prophylactic and routine insertion of a NGT should be avoided because there is weak evidence that gastric decompression decreases anastomotic leakage or ileus, facilitates proceedings, and enhances the recovery of patients [8,9,10,11]. Additionally, a NGT inserted during the surgery should be removed as soon as possible before the reversal of anesthesia [10]. Nevertheless, NGT intubation is occasionally required because of various reasons. For example, some patients need the insertion of a NGT due to the risk of pulmonary aspiration because of gastric insufflation followed by mask ventilation, ileus, or full stomach [15,16,17,23]. In particular, surgical procedures such as pancreaticoduodenectomy often require the insertion of a NGT because of a high incidence of delayed gastric emptying (DGE) [24,25]. According to the report of Lee et al. [24], the overall incidence of DGE was 23.1%, depending on the type of surgical procedure, operation time, amount of intraoperative bleeding or transfusion, etc. Moreover, during the surgical procedure in the Trendelenburg position, bowels in the pelvic cavity can be pulled out by gravity and thereby increase intraabdominal pressure [13,16,26]. Therefore, there exist specific cases in which the NGT must be inserted not only in order to prevent complications, but also in accordance with the needs of the surgical procedure or the patient’s condition [15,16,17,23,24,25]. And the use of a newly designed GDT with a thermistor can be helpful for those patients who need the insertion of NGT by avoiding the double insertion of NGT and esophageal temperature monitoring device which can cause complications such as esophageal damage or malfunction of the NGT.

To determine the insertion length of the NGT, the NEX method, which measures the distance from the tip of the nose to the earlobe to the xiphoid, is frequently used [21]. However, the traditional NEX method has a high risk of malposition in the esophageal danger zone, which leads to an increase in regurgitation of gastric contents due to underestimation [20,27,28]. The tip position of the NGT is ideally between 3 and 10 cm under the lower esophageal sphincter, and side-holes of the tube should be placed below the gastroesophageal junctions (GEJ) to avoid potential risk of complications [20,27]. A previous study reported that the insertion depth of the NG tube needs an additional 9.5 cm to the NEX depth to secure all side-holes passing through the esophagogastric junction [27]. However, the insertion length of NEX + 10 cm can result in tube migration or kinking due to overestimation of the NEX [20]. Thus, there is still no reliable method for determining the appropriate length of an NG tube, and a more adequate method is required. However, it is evident that insertion depth using the NEX method requires an additional depth because of the risks of malposition [20]. In the current study, the averages of NEX depth, actual insertion depth, and calculated optimal insertion depth of GDT were 64.73 ± 6.16 cm, 70.75 ± 5.85 cm, and 70.69 ± 4.66 cm, respectively. Therefore, the optimal insertion depth of this study was about NEX + 5.96 cm, which agreed with the desired insertion depth of the NGT in previous studies [20,27]. 

Likewise, several methods estimate the appropriate depth for the esophageal temperature measurement [18,29]. The appropriate position for the esophageal temperature probe is at the level of the heart to reveal the temperature of the myocardium, which is between T8 and T9 levels or under the tracheal bifurcation [29]. The previous studies estimated it to be approximately 40–45 cm from the nose [3,18,29]. Another study evaluated the distance from the nasal flare to the point between T8 and T9 as 0.228 × standing height − 0.194 [30]. From the above equation, the average insertion depth of an esophageal stethoscope based on the patients’ heights in the current study was 36.45 ± 1.59 cm. Because the thermistor of the GDT is located 35 cm from the end of the tip, the location of the GDT thermistor was calculated as the insertion depth of the GDT − 35 cm. The location of the GDT thermistor according to the actual insertion depth and calculated optimal insertion depth of the GDT were 35.75 ± 5.85 cm and 35.69 ± 4.66 cm, respectively. The GDT thermistor location according to the optimal insertion depth of this study agreed with the calculated esophageal probe depth based on the patient’s height with an average difference of 0.76 ± 3.53 cm.

There are several limitations in the current study. First, the length of the GDT was relatively short for the optimal insertion depth for monitoring esophageal temperature. The average GDT insertion depth for optimal monitoring of esophageal temperature was 61.00 ± 5.40 cm (range 52 to 72 cm). Therefore, in 8 patients, we could not start measuring the esophageal temperature at a depth of NEX + 12 cm, which was longer than 80 cm. And the location of the GDT thermistor was (35.75 ± 5.85 cm) shorter than the conventional location of previous studies (40–45 cm from the nose) or calculated results of 36.45 ± 1.59 cm, which was based on the data of the current study [3,18,29,30]. Therefore, it is considered that the manufacturer needs to modify the GDT to be longer and adjust the location of the thermistor to be deeper. Secondly, the gender difference in our study was 1:3 (male:female), which might have affected the results of the study. Therefore, further evaluation with a large population and research for each gender is required. Finally, we could not estimate the location of the GEJ, which is vital for deciding the insertion depth of the NGT in the CXR [19,20,27]. There is no report about the location of the adult GEJ, but the GEJ in children is located about 25.1 mm below the apex of the left hemidiaphragm in the CXR [31]. However, the location results are from children, not adults, and the distance varies by age [31]. Therefore, we decided to use CLHD for the anatomical landmark instead. Finally, the required insertion depth of the GDT may vary according to the type of operation. Traditionally, the optimal location of the NGT tip is between the GEJ and the pylorus to avoid complications [20,21,27,28]. However, the location of the NGT tip can be moved according to the type of surgery which is associated with the target of the surgical margin. Therefore, the final location of the thermistor can be moved while the depth of the GDT is being adjusted, which can cause inaccurate measurement of core temperature.

## 5. Conclusions

The optimal insertion depth for a GDT with a thermistor was estimated as −15.524 + 0.414 × CCD − 0.145 × CLHD in the patients undergoing laparoscopic surgery in the Trendelenburg position. Estimating the optimal insertion depth of a GDT with a thermistor before anesthesia would help locate the GDT in the appropriate position for both gastric decompression and monitoring of esophageal core temperature.

## Figures and Tables

**Figure 1 ijerph-19-14708-f001:**
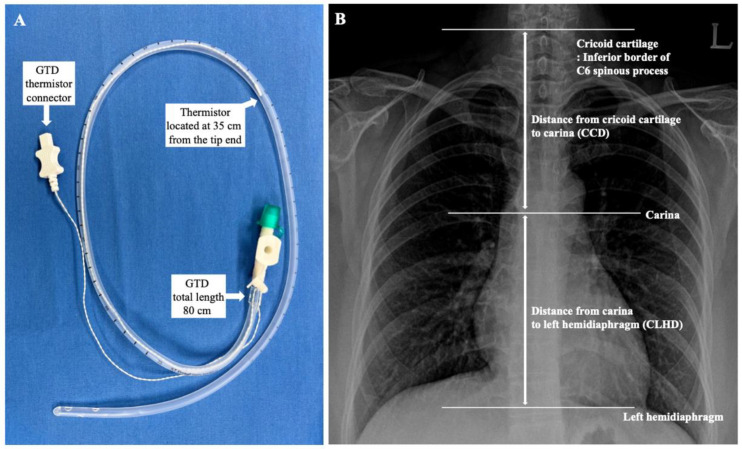
(**A**) Gastric decompression tube (GDT). The total length of the tube was 80 cm, and the thermistor was located at 35 cm from the end of the GDT tip. (**B**) Assessment of anatomical landmarks on chest X-ray.

**Figure 2 ijerph-19-14708-f002:**
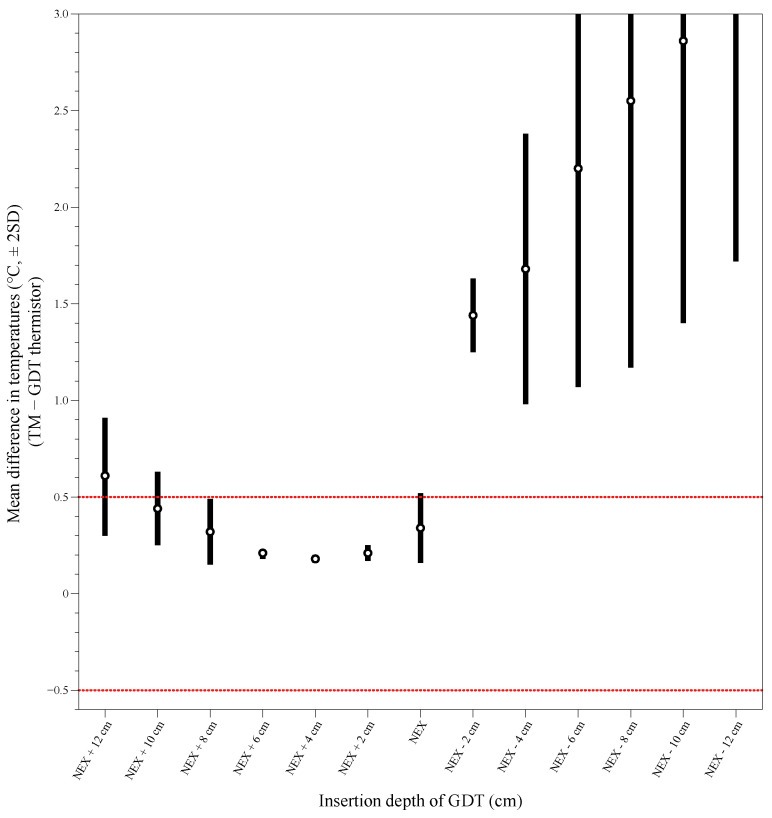
Mean difference ±1.96 standard deviations (SDs) of temperatures between the tympanic membrane and gastric decompression tube (GDT) thermistor. The mean difference ±1.96 SD remained between −0.5 °C and 0.5 °C at the GDT depths within a nose–earlobe–xiphoid distance (NEX) + 8 cm to NEX + 2 cm, which was considered a clinically acceptable location. At NEX + 4 cm, the mean differences in temperature were the most minor between the GDT thermistor and TM.

**Figure 3 ijerph-19-14708-f003:**
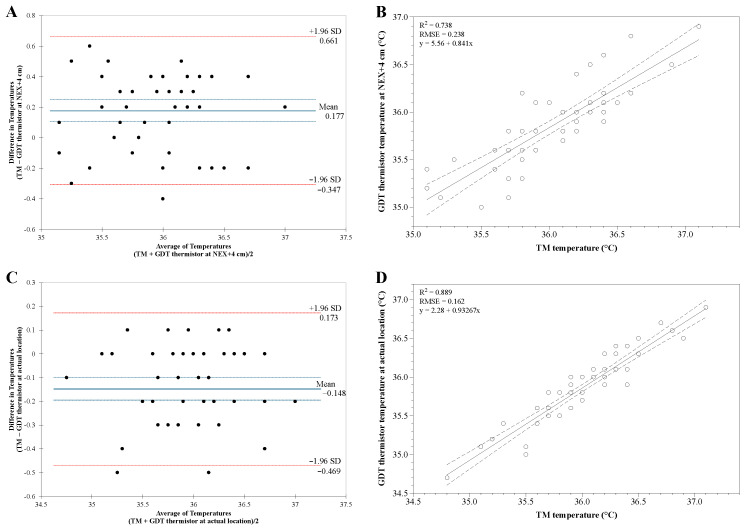
(**A**) Bland–Altman analysis of tympanic membrane (TM) temperature versus gastric decompression tube (GDT) thermistor temperature at a distance of nose–earlobe–xiphoid (NEX) + 4 cm showed good agreement. All points except one were within the clinically acceptable range (±0.5 °C). (**B**) Temperatures of TM and GDT thermistor at NEX + 4 cm showed strong correlation (correlation coefficient 0.862, R^2^ = 0.738). (**C**) Bland–Altman analysis of temperature between TM and GDT thermistor at the actual location with the most minor temperature differences showed better agreement. All points were within the clinically acceptable range (±0.5 °C). (**D**) Temperatures of TM and GDT thermistor at the actual location with the most minor temperature differences showed a stronger correlation (correlation coefficient 0.943, adjusted R^2^ = 0.889). Dotted lines indicate a 95% prediction interval. RMSE, root mean square error. *p* < 0.001.

**Figure 4 ijerph-19-14708-f004:**
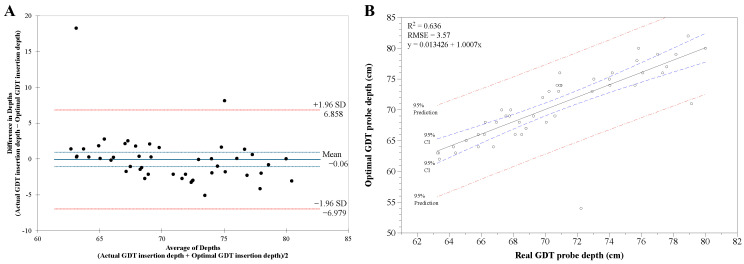
(**A**) Bland-Altman analysis comparing the level of agreement between actual gastric decompression tube (GDT) insertion depth versus optimal GDT insertion depth calculated by using the distances of CDD (cricoid cartilage to carina) and CLHD (carina to left hemidiaphragm). (**B**) Linear regression analysis between the actual GDT insertion depth and optimal GDT insertion depth calculated using CCD and CLHD showed a strong correlation coefficient of 0.877 with adjusted R^2^ = 0.769 and RMSE = 2.630. All values were distributed within the ±1.96 SD and 95% prediction lines except for two values. Blue dotted lines show 95% confidence intervals. Red dotted lines indicate 95% prediction intervals. RMSE, root mean square error. *p* < 0.001.

**Table 1 ijerph-19-14708-t001:** Patient’s characteristics.

Factors	(*n* = 48)
Age (yr)	49.3 (11.0)
Gender (male/female)	12/36
Height (cm)	160.7 (7.0)
Weight (kg)	61.0 (8.9)
BMI (Kg/m^2^)	23.6 (3.1)
ASA class (I/II)	22/26
Coexisting disease	
Hypertension	11
Diabetes	5
Type of surgery	
Laparoscopic hysterectomy	31
Laparoscopic colectomy	17
Distance (mm)	
Cricoid cartilage to carina (CCD)	156.9 (8.1)
Carina to left hemidiaphragm (CLHD)	146.7 (13.8)
Nose–earlobe–xiphoid (NEX)	64.8 (6.2)
Actual insertion depth of GDT	70.8 (5.9)

**Table 2 ijerph-19-14708-t002:** Mean (SD) of GDT probe temperature and differences in temperatures between the tympanic membrane and GDT thermistor according to insertion depths of GDT.

Probe Depth	GDT Probe Temperature (°C)	Difference (TM − GDT Thermistor, °C)	Differences (TM − GDT Thermistor at the Actual Location with the Most Minor Temperature Difference, °C)
NEX + 12 cm (*n* = 35)	35.45 (0.69)	0.61 (0.66)	0.11 (0.16, *n* = 8, 16.7%)
NEX + 10 cm (*n* = 39)	35.62 (0.64)	0.44 (0.51)	0.03 (0.13, *n* = 4, 8.3%)
NEX + 8 cm (*n* = 43)	35.76 (0.58)	0.32 (0.42)	0.22 (0.15, *n* = 5, 10.4%)
NEX + 6 cm (*n* = 45)	35.86 (0.5)	0.21 (0.31)	0.20 (0.15, *n* = 14, 29.2%)
NEX + 4 cm (*n* = 47)	35.9 (0.46)	0.18 (0.25)	0.18 (0.19, *n* = 10, 20.8%)
NEX + 2 cm (*n* = 48)	35.86 (0.51)	0.21 (0.36)	0.10 (0.14, *n* = 5, 10.4%)
NEX (*n* = 48)	35.74 (0.61)	0.34 (0.49)	−0.05 (0.07, *n* = 2, 4.2%)
NEX − 2 cm (*n* = 48)	34.61 (0.65)	1.44 (0.57)	
NEX − 4 cm (*n* = 48)	34.35 (0.91)	1.68 (0.85)	
NEX − 6 cm (*n* = 48)	33.84 (1.16)	2.2 (1.17)	
NEX − 8 cm (*n* = 48)	33.5 (1.28)	2.55 (1.27)	
NEX − 10 cm (*n* = 48)	33.17 (1.32)	2.86 (1.4)	
NEX − 12 cm (*n* = 48)	32.99 (1.25)	3.04 (1.37)	

**Table 3 ijerph-19-14708-t003:** Results of linear regression analysis for actual GDT insertion depth and patient characteristics.

Variables	Univariate Analysis						Multivariate Analysis				
	Correlation Efficient	β	95% CI	*p* Value	Tolerance	VIF	β	95% CI	*p* Value	Tolerance	VIF
Constant							−15.524				
Age (yr)	−0.0.92	0.059	−0.038–0.213	0.268	0.616	1.624					
Gender	0.258	−1.616	−5.931–1.281	0.038	0.471	2.122					
Height (cm)	0.643	0.499	−0.253–1.102	<0.001	0.053	18.936					
Weight (kg)	0.251	−0.552	−1.359–0.399	0.043	0.019	51.657					
BMI (Kg/m^2^)	−0.137	1.548	−1.086–3.667	0.176	0.021	47.44					
CCD (mm)	0.738	0.384	0.218–0.635	<0.001	0.408	2.453	0.414	0.266–0.562	<0.001	0.775	1.290
CLHD (mm)	0.615	0.165	0.058–0.272	<0.001	0.546	1.831	0.145	0.058–0.233	0.002	0.775	1.290

## Data Availability

The analyzed datasets used in this study are available from the corresponding author upon reasonable request.

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
