# Peer review of "Optimal Insertion Depth of Gastric Decompression Tube with a Thermistor for Patients Undergoing Laparoscopic Surgery in Trendelenburg Position"

_ijerph, 2022, doi:10.3390/ijerph192214708_

Round 1
Reviewer 1 Report
Thank you for the opportunity to review this important manuscript. Here are my comments and suggestions.
Line 32-33: this is one of the main points of the study but also the main point for criticism. One recent randomized study concluded that mild hypothermia does not have adverse effects or increase the risk of complications. https://pubmed.ncbi.nlm.nih.gov/35390321/. Please clarify.
Line 43-44. Typically, for elective abdominal operations, a nasogastric tube is not needed. It even increases the risk of respiratory complications with prolonged use. Please do some research on the topic and included important references.
Line 46-47: If the patient did not eat several hours before the operation, there is no risk of aspiration. Otherwise, all patients in general anesthesia would have a nasogastric tube inserted despite primary abdominal pathology as an indication for the operation. Please clarify.
Author Response
Dear Editor and Reviewers.
Thank you for the precious reviews.
I believe your valuable reviews have been of great help to us and will improve the quality of our manuscript.
Reviewer 1:
Line 32-33: this is one of the main points of the study but also the main point for criticism. One recent randomized study concluded that mild hypothermia does not have adverse effects or increase the risk of complications. https://pubmed.ncbi.nlm.nih.gov/35390321/. Please clarify.
: Thank you for the valuable review. I agree with the point that mild hypothermia above 35·5°C is not associated with critical complications. Therefore, avoiding hypothermia, not maintaining normothermia, is essential for the prevention of postoperative complications. We revised the first introduction sentence as below:
Introduction:
General anesthesia impairs the thermoregulatory responses and vasoconstriction threshold, leading to interferences in maintaining core temperature [1]. Moreover, surgical patients are exposed to cold environments [2]. Thus, patients undergoing general anesthesia develop hypothermia easily and have a risk of complications such as coagulation disorder, infection, heart complications, or delayed wound healing [1,3-6]. Therefore, avoiding hypothermia is very important for preventing complications in patients undergoing surgery [2,3,7].
Line 43-44. Typically, for elective abdominal operations, a nasogastric tube is not needed. It even increases the risk of respiratory complications with prolonged use. Please do some research on the topic and included important references.
: We fully agree with you. The answers to this question are included in the answers to the following questions together.
Line 46-47: If the patient did not eat several hours before the operation, there is no risk of aspiration. Otherwise, all patients in general anesthesia would have a nasogastric tube inserted despite primary abdominal pathology as an indication for the operation. Please clarify.
: Your concerns that GDT should not be used routinely, and gastric decompression does not decrease the incidence of respiratory complications are absolutely right, and we agree with that. We also agree that NGT should be used in a limited way. However, sometimes the insertion of GDTs can be essential for some procedures or the patients at risk, we believe that our study can be helpful for the determination of the appropriate depth of GDT. Although we discussed the specific reasons for the insertion of GDT during laparoscopy in the discussion session, we agree that the sentences in the introduction session which claim the need for the GDT insertion are unclear. Therefore, we amended the sentences (previous Lines 43-47) to clarify the need to proceed with this study, the insertion of GDT with a thermistor, and also we added more references to supplement the contents and amended the discussion session, together.
Introduction:
In recent studies, it is known that NGT is not routinely required, does not reduce postoperative complications, and rather increases patient discomfort and pulmonary complications [8-11]. However, using a nasogastric tube (NGT) may sometimes be necessary for several reasons. During various laparoscopic surgery, gastric decompression using an NGT or orogastric tube may be required to avoid intraabdominal organ injury or enhance the surgical view [12-14]. Patients at risk of potential complications such as pulmonary aspiration due to regurgitation of gastric contents may be required gastric decompression during laparoscopic surgery [15,16]. Especially, the gastric decompression tube (GDT) insertion is essential when ileus or full stomach is suspected [17]. Moreover, the Trendelenburg position can increase intraabdominal and intragastric pressure, leading to increased stomach fluid aspiration [13,16]. Therefore, although the GDT should not be inserted as a routine procedure, sometimes it may be required for the surgical procedure or prevention of potential complications.
Discussion:
The insertion of NGT has been used as management in abdominal surgery to facilitate surgical processes by decompressing the distended stomach and enhancing the surgical field's vision [9,12-14]. However, prophylactic and routine insertion of NGT should be avoided because there is weak evidence that gastric decompression decrease anastomotic leakage or ileus, facilitates proceedings, and enhances the recovery of patients [8-11]. And the inserted NGT during the surgery should be removed as soon as possible before the reversal of anesthesia [10]. Nevertheless, NGT intubation is occasionally required because of various reasons. For example, some patients need the insertion of NGT due to the risk of pulmonary aspiration because of gastric insufflation followed by mask ventilation, ileus, or full stomach [15-17,23]. Especially, surgical procedures such as pancreaticoduodenectomy often require the insertion of NGT because of a high incidence of delayed gastric emptying (DGE) [24,25]. According to the report of Lee et al. [24], the overall incidence of DGE was 23.1% which depends on the type of surgical procedure, operation time, amount of intraoperative bleeding or transfusion, etc. Moreover, during the surgical procedure with the Trendelenburg position, bowels in the pelvic cavity can be pulled out by gravity and increase intraabdominal pressure [13,16,26]. Therefore, there exist specific cases in which NGT must be inserted not only in order to prevent complications, but also in accordance with the needs of surgical procedure or patient’s condition [15-17,23-25].
: Thank you again for your precious comments and efforts to improve the quality of our manuscript.
Reviewer 2 Report
Jong et al. performed an interesting study regarding optimal insertion depth of gastric decompression tube during laparoscopic surgery. Basically, this is a well-written paper and merits publication, although the novelty is questioned. In my view, the depth of gastric decompression tube is subjective to the variability of operation types. I suggest the author to further add something in the discussion section focusing this point with consideration for the citation of the data from Asian fellow GI surgeons as followed:
1. Enhanced Recovery After Surgery protocol for elderly Gastric cancer patients: A prospectivestudy for safety and efficacy.
Asian Journal of Surgery, 12 January 2022, ...
Shuo-meng Xiao, Hong-li Ma, … Zhi Ding
2. Ultrasound to guide the individual medical decision by evaluating the gastric contents andrisk of aspiration: A literature review.
Asian Journal of Surgery, 11 March 2020, ...
Gang Zhang, Xiaoyan Huang, … Lan Zhang
3. Is delayed gastric emptying associated with pylorus ring preservation in patientsundergoing pancreaticoduodenectomy?
Asian Journal of Surgery, 18 September 2020, ...
Yun Ho Lee, Young Hoe Hur, … Jin Woong Kim
4. Safety and feasibility of laparoscopic pancreaticoduodenectomy in octogenarians.
Asian Journal of Surgery, 12 October 2021, ...
Ji Su Kim, Munseok Choi, … Chang Moo Kang
Author Response
Dear Editor and Reviewers.
Thank you for the precious reviews.
I believe your valuable reviews have been of great help to us and will improve the quality of our manuscript.
Reviewer 2:
Jong et al. performed an interesting study regarding optimal insertion depth of gastric decompression tube during laparoscopic surgery. Basically, this is a well-written paper and merits publication, although the novelty is questioned. In my view, the depth of gastric decompression tube is subjective to the variability of operation types. I suggest the author to further add something in the discussion section focusing this point with consideration for the citation of the data from Asian fellow GI surgeons as followed:
- Enhanced Recovery After Surgery protocol for elderly Gastric cancer patients: A prospectivestudy for safety and efficacy.
Asian Journal of Surgery, 12 January 2022, ...
Shuo-meng Xiao, Hong-li Ma, … Zhi Ding
- Ultrasound to guide the individual medical decision by evaluating the gastric contents andrisk of aspiration: A literature review.
Asian Journal of Surgery, 11 March 2020, ...
Gang Zhang, Xiaoyan Huang, … Lan Zhang
- Is delayed gastric emptying associated with pylorus ring preservation in patientsundergoing pancreaticoduodenectomy?
Asian Journal of Surgery, 18 September 2020, ...
Yun Ho Lee, Young Hoe Hur, … Jin Woong Kim
- Safety and feasibility of laparoscopic pancreaticoduodenectomy in octogenarians.
Asian Journal of Surgery, 12 October 2021, ...
Ji Su Kim, Munseok Choi, … Chang Moo Kang
: We really thank you for the precious reviews. Your suggestions for the references really helped us to consolidate the hypothesis of our study. Your valuable review improved our manuscript.
: Your suggested references are inserted into the introduction and discussion sessions.
: We revised the discussion session and added some limitations as per your suggestions.
Discussion:
The insertion of NGT has been used as management in abdominal surgery to facilitate surgical processes by decompressing the distended stomach and enhancing the surgical field's vision [9,12-14]. However, prophylactic and routine insertion of NGT should be avoided because there is weak evidence that gastric decompression decrease anastomotic leakage or ileus, facilitates proceedings, and enhances the recovery of patients [8-11]. And the inserted NGT during the surgery should be removed as soon as possible before the reversal of anesthesia [10]. Nevertheless, NGT intubation is occasionally required because of various reasons. Some patients need the insertion of NGT due to the risk of pulmonary aspiration because of gastric insufflation followed by mask ventilation, ileus, or full stomach [15-17,23]. Especially, surgical procedures such as pancreaticoduodenectomy often require the insertion of NGT because of a high incidence of delayed gastric emptying (DGE) [24,25]. According to the report of Lee et al. [24], the overall incidence of DGE was 23.1% which depends on the type of surgical procedure, operation time, amount of intraoperative bleeding or transfusion, etc. Moreover, during surgical procedure with Trendelenburg position, bowels in the pelvic cavity can be pulled out by gravity and increase intraabdominal pressure [13,16,26]. Therefore, there exist specific cases in which NGT must be inserted not only in order to prevent complications, but also in accordance with needs of surgical procedure or patient’s condition [15-17,23-25].
Discussion – limitations:
Finally, the required insertion depth of GDT may vary according to the types of operation. Traditionally, the optimal location of the NGT tip is between GEJ and pylorus to avoid complications [20,21,27,28]. However, the location of the NGT tip can be moved according to the types of surgery which is associated with the target of the surgical margin. Therefore, the final location of the thermistor can be moved while the adjustment of depth of GDT, which can cause the inaccurate measurement of core temperature.
: Thank you again for your efforts to review and advise our manuscript.